# SARS-CoV-2 infection- induced seroprevalence among children and associated risk factors during the pre- and omicron-dominant wave, from January 2021 through December 2022, Thailand: A longitudinal study

Nungruthai Suntronwong[1], Preeyaporn Vichaiwattana[1], Sirapa Klinfueng[1], Jiratchaya Puenpa[1], Sitthichai Kanokudom[1,2], Suvichada Assawakosri[1,2], Jira Chansaenroj[1], Donchida Srimuan[1], Thaksaporn Thatsanatorn[1], Siriporn Songtaisarana[1], Natthinee Sudhinaraset[1], Nasamon Wanlapakorn[1], Yong Poovorawan[1,3]*

1 Center of Excellence in Clinical Virology, Faculty of Medicine, Chulalongkorn University, Bangkok, Thailand, 2 Center of Excellence in Osteoarthritis and Musculoskeleton, Faculty of Medicine, Chulalongkorn University, King Chulalongkorn Memorial Hospital, Thai Red Cross Society, Bangkok, Thailand, 3 The Royal Society of Thailand (FRS(T)), Sanam Sueapa, Dusit, Bangkok, Thailand

* yong.p@chula.ac.th

## Abstract

### Background

Severe acute respiratory syndrome-coronavirus-2 (SARS-CoV-2) infection can be asymptomatic in young children. Therefore, the true rate of infection is likely underestimated. Few data are available on the rate of infections in young children, and studies on SARS-CoV-2 seroprevalence among children during the omicron wave are limited. We assessed the SARS-CoV-2 infection-induced seroprevalence among children and estimated the associated risk factors for seropositivity.

### Methods

A longitudinal serological survey was conducted from January 2021 through December 2022. The inclusion criteria were healthy children between 5 and 7 years old and their parents or legal guardians provided written informed consent. Samples were tested for anti-nucleocapsid (N) IgG and anti-receptor binding domain (RBD) IgG using a chemiluminescent microparticle immunoassay (CMIA), and total anti-RBD immunoglobulin (Ig) was detected using an electrochemiluminescence immunoassay (ECLIA). The vaccination and SARS-CoV-2 infection history were collected.

### Results

In all, 457 serum samples were obtained from 241 annually followed-up children in this longitudinal serological survey. Of these, 201 participants provided samples at two serial time

**Data Availability Statement:** All relevant data are within the paper and its Supporting Information files.

**Funding:** This work was supported by the National Research Council of Thailand, the Health Systems Research Institute, the Center of Excellence in Clinical Virology of Chulalongkorn University, King Chulalongkorn Memorial Hospital, and the Berli Jucker Company Big C foundation. Nungruthai Suntronwong reports that financial support was also provided by the Second Century Fund Fellowship of Chulalongkorn University. The funders had no role in study design, data collection and analysis, decision to publish, or preparation of the manuscript.

**Competing interests:** The authors have declared that no competing interests exist.

points—during the pre-omicron and omicron-dominant wave. Overall, seroprevalence induced by SARS-CoV-2 infection increased from 9.1% (22/241) during the pre-omicron to 48.8% (98/201) during the omicron wave. Amongst seropositive individuals, the infection-induced seropositivity was lower in vaccinated participants with two doses of BNT162b2 than in the unvaccinated participants (26.4% vs. 56%; OR, 0.28; 95%CI: 0.14–0.58). Nevertheless, the ratio of seropositive cases per recalled infection was 1.63 during the omicron dominant wave. The overall seroprevalence induced by infection, vaccination, and hybrid immunity was 77.1% (155/201) between January and December 2022.

## Conclusions

We report an increase in infection-induced seroprevalence among children during the omicron wave. These findings highlight that a seroprevalence survey can help determine the true rate of infection, particularly in asymptomatic infection, and optimize public health policies and vaccine strategies in the pediatric population.

## Introduction

Most children with severe acute respiratory syndrome-coronavirus-2 (SARS-CoV-2) infection are asymptomatic at presentation. Thus, mildly symptomatic and asymptomatic children might not be tested for or diagnosed with SARS-CoV-2 infection [1]. As a result, the incidence of pediatric coronavirus disease 2019 (COVID-19) cases is underestimated and likely higher than reported according to the reverse transcriptase-polymerase chain reaction (RT-PCR) results. Furthermore, children are commonly infected after close contact with other members in the same household. They are also very likely unaware of their infection status, potentially contributing to virus transmission [2].

Although children with COVID-19 typically develop a milder illness than adults, long-term complications including multisystem inflammatory syndrome in children (MIS-C) can occur after SARS-CoV-2 infection [3]. Recent evidence notes that COVID-19 vaccination in children is associated with a high level of protection against MIS-C and a reduced rate of hospitalization and deaths [4]. In Thailand, the Thai Food and Drugs Administration have approved BNT162b2, an mRNA vaccine, for vaccination in children aged 5–11 years old, and Corona-Vac and BBIBP-CorV, which are inactivated vaccines, for children aged ≥6 years old. Consequently, data on seroprevalence in children is essential to inform and guide vaccine policies.

A serological survey provides data regarding the retrospective detection of individuals who had a previous infection, even in an asymptomatic case, and can help monitor virus transmission [5]. Anti-nucleocapsid (N) and anti-receptor binding domain (RBD) antibodies are elicited after natural infection or vaccination. However, while anti-N antibodies can be detected after receipt of an inactivated vaccine, they are not detected in individuals who are vaccinated with the viral vector or mRNA vaccines. Therefore, Thai children vaccinated with an inactivated vaccine could elicit the anti-N IgG response even without previous SARS-CoV-2 infection. Consequently, the detection of anti-N and anti-RBD antibodies in unvaccinated individuals and anti-N antibodies in vaccinated individuals with mRNA vaccines implies natural infection-induced immunity. In addition, detection of anti-RBD IgG can provide information about the population-based immunity induced by natural infection, vaccination, or both. A previous study showed that individuals with a high level of anti-RBD IgG are associated with

developing an effective immune response against the virus, as indicated by a high titer of neutralizing antibodies [6].

The omicron variant was first detected in Thailand in mid-December 2021. The incidence of omicron infection rapidly increased and subsequently became the predominant variant by January 2022 [7]. A study demonstrated that the omicron variant was highly transmissible and can evade the immune system even in vaccinated or previously infected individuals [8]. Consistent with this finding, the spreading of omicron variants peaked between January and March 2022. In addition, infection by the omicron variant resulted in milder symptoms and was associated with lower risks of hospitalization and death than the delta variant [9]. Population-based surveillance indicated that the COVID-19-associated hospitalization rates of children aged 5–11 years was 2.8 per 100,000 children during the omicron-predominant peak [10]. Longitudinal serological surveillance in Uganda showed that 84.8% unvaccinated individuals were infected for the first time and 50.8% individuals were re-infected during the omicron wave [11]. Additionally, an increase of infection-induced seroprevalence during the omicron wave has been reported in South Africa [12]. In the United States, serological studies reported that more than half the children were infected with the omicron variant [13]. However, few data are available on the rate of infections in young children, and studies on the SARS-CoV-2 seroprevalence among children during the omicron wave are still limited.

In this study, we assessed the seroprevalence of anti-SARS-CoV-2 antibodies induced by SARS-CoV-2 infection from January 2021 to December 2022 (the duration spanned the pre- and omicron wave) and estimated the associated risk factors for seropositivity. Data on seroprevalence can help to estimate the number of children who experience SARS-CoV-2 infection and assess the impact of immunization to protect from SARS-CoV-2 infection in the pediatric population, which is paramount to establish public health prevention and vaccination strategies.

## Materials and methods

### Study design and sample collection

This study analyzed the serum samples from children who were followed-up in the longitudinal serological study of pertussis vaccine immunity at the Center of Excellence in Clinical Virology, Chulalongkorn Memorial Hospital, Bangkok, Thailand, from January 1, 2021 through December 14, 2022 [14]. Participants were invited to complete surveys on SARS-CoV-2 infection and COVID-19 vaccination. This survey included questions regarding previous SARS-CoV-2 infections detected by RT-PCR test or antigen test kit (ATK); date of infection; and vaccination data including type, dose, and date of vaccination, as reported by the children's parents. The survey responses were retrospectively collected between May and December 2022. The inclusion criteria were healthy children who were followed-up in an immunogenicity cohort study between 2015 and 2022 [14]. Children aged between 5 and 7 years old and whose parents were able to provide written informed consent were enrolled. The exclusion criteria were parents who did not want to disclose the vaccination status and infection history and children who previously received any dose of the inactivated COVID-19 vaccine, because such individuals could elicit the anti-N IgG response from the inactivated vaccine.

The Research Ethics Committee of the Faculty of Medicine, Chulalongkorn University, approved the study (IRB number: 173/63). The study protocol adhered to the tenets of the Declaration of Helsinki and Good Clinical Practice principles. Written informed consent was obtained from the parents or legal guardians of all participating children.

## Serological analysis

Sera samples collected from eligible participants were subjected to IgG-specific testing against SARS-CoV-2 nucleocapsid (N) protein using a chemiluminescent microparticle immunoassay (CMIA) (Abbott Architect Immunoassay; Abbott Diagnostics, Abbott Park, IL, USA). According to the manufacturer's instructions, we considered samples with a signal-to-cut-off ratio ≥1.4 as anti-N IgG positive, while samples with a value <1.4 were considered as seronegative. The total immunoglobulin (Ig) and IgG specific against the SARS-CoV-2 receptor binding domain (RBD) were quantitatively measured using the Roche Elecsys anti-SARS-CoV-2 immunoglobulin immune assay (Roche) and SARS-CoV-2 Quant IgG II (Abbott Diagnostics, Abbott Park, IL, USA), respectively. Antibody level is expressed as U/mL for anti-RBD Ig, with a cut-off value ≥0.8 defined as positive and BAU/mL for anti-RBD IgG, with values ≥7.1 defined as positive.

## Classification of samples based on seropositivity

Infection-induced seropositivity was estimated based on the presence of anti-RBD Ig (≥0.8 U/mL), anti-RBD IgG (≥7.1 BAU/mL), or anti-N IgG (≥1.4 S/C) among unvaccinated children and the presence of anti-N IgG in vaccinated individuals with BNT162b2 vaccine. Vaccination status was classified according to the dates of reported infection and vaccination. Overall seroprevalence induced by infection, vaccination, and hybrid immunity was estimated based on the anti-RBD antibody seropositivity in both unvaccinated and vaccinated participants.

## Statistical analysis

Seroprevalence of SARS-CoV-2 was defined as the number of children who tested seropositive divided by the total number of children tested and is presented on monthly and quarterly seropositive rates in Figs 2 and 3. Infection-induced seropositivity associated with potential risk factors was estimated based on odd ratio (OR) which was calculated using McNemar's test for paired serum samples (comparing between the pre-omicron and omicron-dominant wave) and chi-square test for independent samples. The ratio of children with infection-induced seropositivity during the omicron-dominant wave per recalled of previous SARS-CoV-2 infection were assessed. Statistical analysis was conducted using SPSS v23.0 (IBM Corp, Armonk, NY, USA). A *p*-value <0.05 was considered to indicate statistically significant differences.

# Results

## Study participants

In all, 457 samples from 241 enrolled participants at the source study were included (Fig 1). Of these, 241 samples were collected from January through December 2021 (hereafter referred to as the pre-omicron wave), and 216 samples were collected from January to December 2022 (hereafter referred to as the omicron-dominant wave). Between January and December 2022, two and 13 participants were excluded because of lacking vaccination history and receipt of an inactivated vaccine, respectively. Thus, 442 serum samples from 241 participants were eligible and met the criteria for the final analysis.

In total, 442 samples were obtained, including 54.5% (241/442) serum samples collected between January and December 2021 and 45.5% (201/442) collected between January and December 2022, during which time omicron was predominantly circulating. The mean (SD) age of study participants enrolled during the pre-omicron and omicron-dominant wave was 5.25 (0.4) and 6.2 (0.4) years old, respectively, and there were 51% female and 49% male subjects (S1 Table). Of these, 201 participants provided samples at two serial time points,

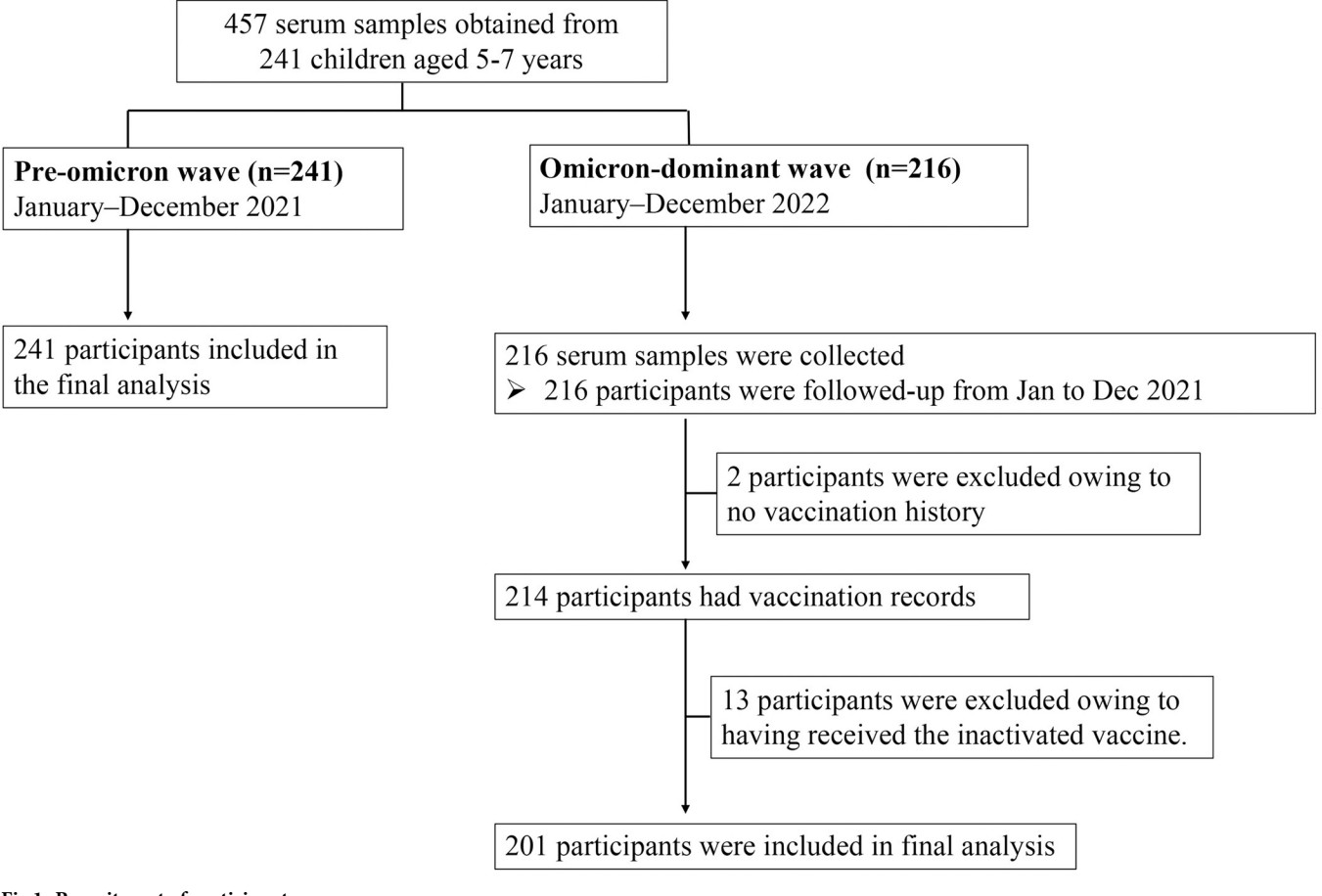

**Fig 1. Recruitment of participants.**

including during the pre- and omicron-dominant wave, and the median interval between the two blood samplings was 362 days (IQR: 343–376). During the omicron-dominant wave, 45.8% (92/201) participants received at least one dose of BNT162b2 by February 2022 including 40.2% (37/92), 57.6% (53/92), and 2.2% (2/92) participants who had received a single dose, two-doses, and three doses of BNT162b2, respectively. No participant enrolled during the pre-omicron wave received the COVID-19 vaccine.

## Seroprevalence induced by SARS-CoV-2 infection

There were 120 samples out of a total of 442 that tested anti-RBD Ig or anti-N IgG seropositive for unvaccinated participants and anti-N IgG seropositive or having reported of SARS-CoV-2 infection for vaccinated participants, resulting in an estimated infection-induced seropositivity of 27.1% (120/442) (Fig 2). The infection-induced seroprevalence increased from 9.1% (22/241) between January and December 2021 to 48.8% (98/201) between January and December 2022 (S2 Table). In 2022, the increasing of infection-induced seropositivity was observed by March 2022 and reached 62% (31/50) and 63.9% (23/36) by the third and fourth quarters of 2022, respectively (Fig 2). Amongst seropositive participants during the pre-omicron wave (n = 22), 17 were followed-up for the serological cohorts during the omicron-dominant wave, and 88.2% (15/17) children still tested anti-N IgG seropositive and provided a history of infection during the omicron-dominant wave.

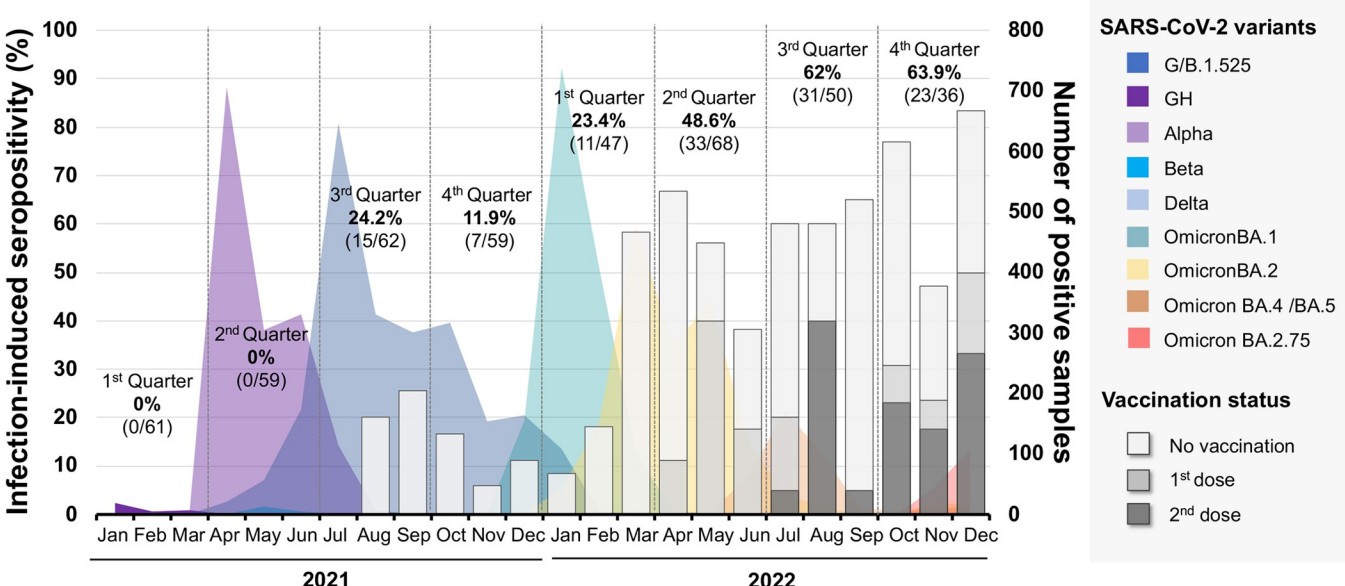

**Fig 2. Monthly infection-induced seroprevalence in children and distribution of SARS-CoV-2 variants between January 2021 and December 2022.** A total of 442 sera samples collected from Thai children were used to determine the presence of SARS-CoV-2 antibodies to estimate SARS-CoV-2 infection-induced seropositivity. SARS-CoV-2 infection-induced seropositivity was defined as the seropositivity of anti-N IgG or anti-RBD antibody for unvaccinated individuals and anti-N IgG seropositivity or having reported SARS-CoV-2 infection for vaccinated individuals with the BNT162b2 vaccine. The stacked bar graph represents the percentage of infection-induced seropositivity according to the vaccination status (left Y-axis). The vertical dotted lines indicate the quarter, and the number above the bar graph indicates infection-induced seropositivity estimated for each quarter. Color shaded areas indicate the number of nasopharyngeal swab samples that tested positive (right Y axis) and subsequently identified the SARS-CoV-2 variants (n = 7,729) that was obtained from our previous study [7]. Data on SARS-CoV-2 variants were identified using sequencing and multiplex real-time reverse transcription polymerase chain reaction (RT-PCR) to depict the predominant wave for individual SARS-CoV-2 variants in Thailand.

### Factors associated with infection-induced seropositivity

Among unvaccinated participants, samples collected during the omicron dominant wave were more likely seropositive than those collected during the pre-omicron wave (53.5% vs. 9.1%; OR = 50.0; 95%CI: 8.6–2013.9, $p$ <0.001) (Table 1). Notably, vaccinated participants with two-doses of BNT162b2 were less likely infected with SARS-CoV-2 than unvaccinated participants (26.4% vs. 56.0%; OR = 0.28; 95%CI: 0.14–0.58; $p$ <0.001). Infection-induced seropositivity was not significantly different between participants who were immunized with a single dose of BNT162b2 and those who were unvaccinated. Moreover, there was no difference in seropositive rate between participants living in a household with ≥5 members and those living in a household with <5 members during the omicron wave. However, children living with household members who had previous COVID-19 infection had a higher risk of seropositivity than those living with household members without SARS-CoV-2 infection (69.1% vs. 22.9%; OR = 11.3; 95%CI: 5.3–23.9; $p$ <0.001).

### Infection-induced seropositivity associated with previous SARS-CoV-2 testing

We further estimated the ratio of infection-induced seropositive cases per recalled infection during the omicron-dominant wave. Overall, 90.5% (182/201) participants completed a survey on infection history. Subsequently, 93 of 182 children showed infection-induced seropositivity. Of these, 61.3% (57/93) previously reported SARS-CoV-2 infection, and the median interval between dates of infection and blood collection was 94 days (IQR: 64–168.5), while 38.7% (36/

**Table 1. SARS-CoV-2 infection-induced seropositivity and associated potential risk factors.**

| Characteristic | Overall | Infection-induced seroprevalence | | Odd ratio (OR) |
|---|---|---|---|---|
| | n | n | (%) | (95%CI) |
| Overall | | | | - |
| • Pre-omicron wave | 241 | 22 | 9.1% | Ref [a] |
| • Omicron dominant wave | 201 | 98 | 48.8% | 41.5 (11.13–348.37)*** |
| Sex, n (%) according to the study period | | | | |
| Pre-omicron wave | | | | |
| Boy | 118 | 12 | 10.2% | Ref [b] |
| Girl | 123 | 10 | 8.1% | 0.78 (0.32–1.88) |
| Omicron-dominant wave | | | | |
| Boy | 98 | 46 | 46.9% | Ref [b] |
| Girl | 103 | 52 | 50.5% | 1.15 (0.66–2.00) |
| Vaccination status according to dates of infection and vaccination | | | | |
| Unvaccinated | | | | |
| Pre-omicron wave | 241 | 22 | 9.1% | Ref [a] |
| Omicron-dominant wave | 109 | 61 | 53.5% | 50.0 (8.6–2013.9) *** |
| Omicron dominant wave (n = 201) | | | | |
| Unvaccinated | 109 | 61 | 56.0% | Ref [b] |
| Vaccination | | | | |
| -BNT162b2 (1 dose) | 37 | 23 | 62.2% | 1.29 (0.60–2.78) |
| -BNT162b2 (2 doses) | 53 | 14 | 26.4% | 0.28 (0.14–0.58)*** |
| -BNT162b2 (3 doses) | 2 | 0 | 0% | - |
| Pre-omicron wave Number of household members, n | | | | |
| 2–4 | 99 | 5 | 5.1% | Ref [b] |
| ≥5 | 69 | 12 | 17.4% | 3.96 (1.33–11.81)** |
| N/A | 73 | 5 | 6.8% | - |
| Omicron-dominant wave Number of household members, n | | | | |
| 2–4 | 99 | 44 | 44.4% | Ref [b] |
| ≥5 | 69 | 40 | 58.0% | 1.72 (0.93–3.21) |
| N/A | 33 | 14 | 42.4% | - |
| Confirmed COVID-19 infection in household members tested via PCR or ATK during the omicron wave, n | | | | |
| No | 70 | 16 | 22.9% | Ref [b] |
| Yes | 97 | 67 | 69.1% | 11.3 (5.3–23.9)*** |
| N/A | 34 | 15 | 44.1% | - |

[a] Odd ratio (OR) was calculated based on pair-serum samples and statistical analysis was performed using the McNemar's test.

[b] Odd ratio (OR) was calculated based on independent samples by using the chi-square test.

* indicates p<0.05

***indicates p<0.01***indicates p<0.001.

93) reported being unaware of their infection status (Table 2). Therefore, the ratio of infection-induced seropositive cases per recalled infection during the omicron-dominant wave was 1.63 (93/57). Moreover, only 10.1% (9/89) seronegative participants reported previous SARS-CoV-2 infection, for which the median interval between infection and blood collection dates was 260 days (IQR: 182–310).

**Table 2. Association between infection-induced seropositivity and previous SARS-CoV-2 infection.** Number of children who complete a survey of infection history (n = 182) and infection-induced seropositivity during the omicron-dominant wave (January–December 2022).

|  | Infection-induced seropositive [b] (n = 93) | Infection-induced seronegative (n = 89) |
|---|---|---|
| History of previous infection [a] (n = 66) | 57 | 9 |
| Interval between the reported date of infection and blood collection, median (IQR) | 94 (64–168.5) | 260 (182–310) |
| No history of previous infection (n = 116) | 36 | 80 |

[a] Previous SARS-CoV-2 infection was diagnosed by RT-PCR or ATK test

[b] Infection-induced seropositivity was estimated based on the presence of anti-RBD Ig ($\geq$ 0.8 U/mL), anti-RBD IgG ($\geq$7.1 BAU/mL) or anti-N IgG ($\geq$ 1.4 S/C) among unvaccinated children and the presence of anti-N IgG in vaccinated individuals with BNT162b2 vaccine.

## Seroprevalence induced by infection, vaccination, and hybrid immunity

We further estimated the overall seroprevalence induced by natural infection, vaccination, and hybrid immunity based on the seropositivity of anti-RBD antibodies (Fig 3). Regarding the vaccine implementation in children, the seropositivity of anti-RBD antibodies was 77.1% (155/201) between January and December 2022, which increased by March 2022 and was >80% (95%CI: 85–96%) from April 2022.

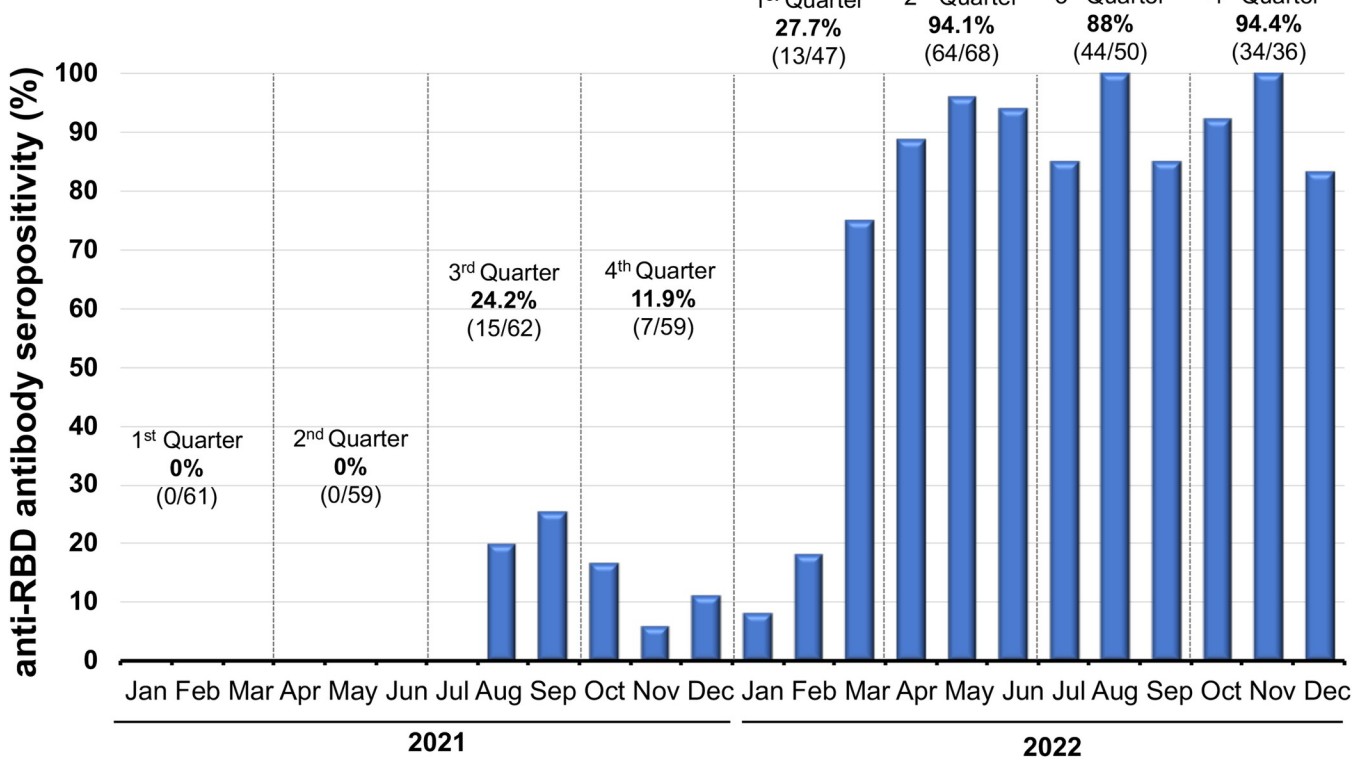

**Fig 3. Monthly distribution of seroprevalence induced by infection, vaccination, and hybrid immunity observed in children aged 5–7 years old between January 2021 and December 2022.** Seroprevalence induced by infection, vaccination, and hybrid immunity was estimated based on the anti-RBD antibody seropositivity in both unvaccinated and vaccinated participants.

## Discussion

Our findings report an increase in infection-induced seroprevalence among children from January through December 2022, indicating the high rates of SARS-CoV-2 infection during the omicron wave. This is in agreement with other studies that reported a significant rise in the seroprevalence rate among children during the omicron wave in South Africa [12] and the United States [13]. This finding supported a previous report that showed that the omicron variant is more transmissible than the previous SARS-CoV-2 variants [8].

Our results also showed that even children who received the BNT162b2 vaccine were anti-N IgG seropositive during the omicron wave. This result might be related to antibody evasion by the omicron variants, and the limit of vaccine efficacy among children [15]. Although vaccinated children were susceptible to being infected by the omicron variant, the risks of infection and hospitalization were reduced for vaccinated compared to unvaccinated children aged between 5 and 11 years [16]. In addition, our study showed that vaccinated children who received two doses were less likely to be seropositive than those who received only a single dose. This result was supported by a meta-analysis study that suggested that the estimated mean secondary attack rates observed during the omicron wave were higher in partially vaccinated cases (76.8%, 95%CI: 7.7%–99.2%) than in fully vaccinated cases (50.8%, 95%CI: 47.9%–53.8%) [17]. Considering the highly contagious new SARS-CoV-2 variants; however, an update or more effective COVID-19 vaccine against a wide range of variants is needed [18]. To date, the U.S. Food and Drug Administration amended the updated bivalent COVID-19 vaccine containing the original and omicron variant and has been approved for children down to 6 months of age [19].

Although an increased number of household members did not affect the risk of seropositivity in children, we found that a higher risk of seropositivity was observed in children living with infected household members than those living with non-infected household members. This finding was consistent with a study from Switzerland that indicated the risk of being seropositive in children aged <6 years was more likely associated with the number of household members who tested positive for SARS-CoV-2, indicating the SARS-CoV-2 transmission for children aged <6 years occurs commonly within the family [20]. In addition, our study was consistent with previous reports in Uganda that reported no significant difference in infection-induced seropositivity based on the gender of the study participants [11]. This was not consistent with a previous report that showed that adult females were more likely to be seropositive than adult males [12].

The proportion of anti-N IgG or anti-RBD antibody seropositivity among unvaccinated and anti-N IgG seropositivity among vaccinated cases may reflect the true rate of infections, including mildly symptomatic or asymptomatic cases and undetected by COVID-19 testing. Our study found that 38.7% children aged 5–7 years with seropositivity reported no previous infection. This result may reflect either asymptomatic infection or undetected infection upon COVID-19 testing in patients likely unaware of their infection status, in which case the virus remains actively transmissible [21]. In addition, our estimated ratio of seropositive cases per recall infection was lower than that reported in a seroprevalence study in children aged 3–11 years from Germany between March and May 2021 (1.63 vs. 3.0) [22] and another study of awareness in adults during the omicron wave (1.63 vs. 2.28) [5].

The anti-N IgG seronegative results in children that reported infection may be owing to the long interval from infection to blood sampling and the severity of disease after infection. This result was supported by a previous study in adults indicating that 87.5% individuals had detectable anti-N IgG even 3 months post-infection, while only 26.6% individuals had detectable the anti-N IgG 12 months post-infection [23]. In addition, recent evidence showed that the

sensitivity of anti-N IgG detection among individuals with recent infection defined by RT-PCR testing ranged from 74% to 81% during the omicron wave [24]. To overcome the limitation of anti-N IgG detection, our study included anti-RBD antibodies to classify the seropositivity induced by SARS-CoV-2 infection in unvaccinated participants and considered evidence of SARS-CoV-2 infection in vaccinated participants.

The limitations of the study include the small sample size, as this may not represent the seroprevalence in the general population. Further, the present study could have underestimated the cumulative seropositivity because children with inactivated vaccines were not included. Furthermore, it is important to note that the anti-N antibody titers tend to decrease over time, and the sensitivity of the anti-N IgG assay to detect the antibody may be influenced by factors such as the time interval between the infection and blood collection, as well as the individual's vaccination status [25]. Consequently, when interpreting the results, it is crucial to consider the possibility of false positive or false negative outcomes, as well as the potential failure to develop detectable antibodies following vaccination or infection.

In conclusion, our study reports that the infection-induced seropositivity was increased among children because of the spread of the omicron variant which emphasizes the need for updating or more effective COVID-19 vaccine. By illustrating a surge in infection-induced seropositive in Thailand during the omicron wave, the data allowed for a more accurate estimation of the actual number of children who experienced COVID-19. This finding showed the benefit of anti-SARS-CoV-2 antibody detection in assessing the seroprevalence induced by infection, which can subsequently track the transmission of SARS-CoV-2 variants and determine the impact of immunization on the pediatric population.

## Supporting information

**S1 Table. Baseline characteristics and data summaries of vaccination status and infection history of final study participants recruited from January to December 2021 (pre-omicron wave) and January to December 2022 (omicron-dominant wave).** Sera samples were categized according to the dates of blood collection into pre- and omicron dominant wave. (DOCX)

**S2 Table. Number of SARS-CoV-2 infection-induced seropositive and seronegative per month detected in children aged 5–7 years old between pre- (January-December 2021) and omicron dominant wave (January-December 2022).** (DOCX)

## Acknowledgments

We would like to thank all the staff of the Center of Excellence in Clinical Virology and all the participants for helping and supporting in this project.

## Author Contributions

**Conceptualization:** Nungruthai Suntronwong, Nasamon Wanlapakorn, Yong Poovorawan.

**Data curation:** Nungruthai Suntronwong, Jiratchaya Puenpa, Donchida Srimuan, Thaksaporn Thatsanatorn, Siriporn Songtaisarana, Natthinee Sudhinaraset.

**Formal analysis:** Nungruthai Suntronwong.

**Methodology:** Preeyaporn Vichaiwattana, Sirapa Klinfueng, Sitthichai Kanokudom, Suvichada Assawakosri, Jira Chansaenroj.

**Project administration:** Yong Poovorawan.

**Writing – original draft:** Nungruthai Suntronwong.

**Writing – review & editing:** Nungruthai Suntronwong, Sitthichai Kanokudom, Nasamon Wanlapakorn, Yong Poovorawan.

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
