## [Decision Letter · Decision Letter 0]

15 Feb 2023

PONE-D-22-32927SARS-CoV-2 infection- induced seroprevalence among children and associated risk factors during pre- and omicron-dominant wave, from January 2021 through November 2022, Thailand: Longitudinal studyPLOS ONE

Dear Dr. Poovorawan,

Thank you for submitting your manuscript to PLOS ONE. After careful consideration, we feel that it has merit but does not fully meet PLOS ONE’s publication criteria as it currently stands. Therefore, we invite you to submit a revised version of the manuscript that addresses the points raised during the review process.

We look forward to receiving your revised manuscript.

Kind regards,

Harapan Harapan, MD, PhD

Academic Editor

PLOS ONE

Journal Requirements:

“This research was financially supported by the Health Systems Research Institute (HSRI), National Research Council of Thailand (NRCT), the Center of Excellence in Clinical Virology, Chulalongkorn University, and King Chulalongkorn Memorial Hospital, MK Restaurant Group Aunt Thongkam Foundation, BJC Big C Foundation, and the Second Century Fund (C2F), Chulalongkorn University”

“This work was supported by the National Research Council of Thailand, the Health Systems Research Institute, the Center of Excellence in Clinical Virology of Chulalongkorn University, King Chulalongkorn Memorial Hospital, and the Berli Jucker Company Big C foundation. Nungruthai Suntronwong reports that financial support was also provided by the Second Century Fund Fellowship of Chulalongkorn University. The funders had no role in study design, data collection and analysis, decision to publish, or preparation of the manuscript.”

Reviewers' comments:

Reviewer's Responses to Questions

**Comments to the Author**

1. Is the manuscript technically sound, and do the data support the conclusions?

Reviewer #1: Partly

Reviewer #2: Partly

Reviewer #3: Partly

Reviewer #4: Yes

2. Has the statistical analysis been performed appropriately and rigorously? 

Reviewer #1: Yes

Reviewer #2: No

Reviewer #3: No

Reviewer #4: No

3. Have the authors made all data underlying the findings in their manuscript fully available?

Reviewer #1: Yes

Reviewer #2: Yes

Reviewer #3: Yes

Reviewer #4: No

4. Is the manuscript presented in an intelligible fashion and written in standard English?

Reviewer #1: Yes

Reviewer #2: Yes

Reviewer #3: Yes

Reviewer #4: No

5. Review Comments to the Author

Reviewer #1: This study by Suntronwong et al describes “SARS-CoV-2 infection- induced seroprevalence among children and associated risk factors during pre- and omicron-dominant wave, from January 2021 through November 2022, Thailand: Longitudinal study”. It will be nice study if the authors will consider few points and answer it properly:

Abstract

1. At end of abstract, authors should emphasis on novelty and impact of the study.

Introduction:

I. What was the criterion for selection of pediatric population?

I. Is any similar type of data was reported from any part of the world. If yes, then please include in introduction?

II. Authors are also suggested to cite the some of the recently published papers regarding the longitudinal study regarding SARS-CoV-2 infection- induced seroprevalence.

Result and discussion

• Why the samples were taken from Center of Excellence in Clinical Virology, Chulalongkorn Memorial Hospital, Bangkok, Thailand.

• What was the control for this study?

• Why these particular samples were taken?

• What is the importance of this study as the samples were taken from specific area?

• Why the samples were taken in between January and December?

• As the time duration is too long where there is possibility of change in sero-positivity.

Conclusion

Please add impact of study on the national level.

Reviewer #2: 1. Abstract. Briefly present the inclusion/exclusion criteria in the ‘method’ part. Children age range should be disclose too.

2. In introduction. Authors are encouraged to provide the Omicron prevalence among children.

3. In introduction. Please include the necessity of performing this study especially in the midst of global emergence of Omicron VOC (https://www.sciencedirect.com/science/article/pii/S1876034122003161?via%3Dihub#fig0010)

4. Please explain the impact of low seroconversion among CoronaVac recipients to the results of your study. (Ref: https://www.narraj.org/main/article/view/71)

5. High population burden of Omicron variant (B.1.1.529) has been associated with the emergence of severe hepatitis of unknown etiology in children (ref: https://doi.org/10.1016/j.ijid.2022.05.028) How this finding is relevant to your study?

6. There is a good report titled “The race for clinical trials on Omicron-based COVID-19 vaccine candidates: Updates from global databases” (https://www.narraj.org/main/article/view/88) which authors may wish to incorporate in their discussion.

7. It is interesting that there is no meaningful difference on the infection-induced seroprevalence between boy and girl. Meanwhile, gender has been an associated factor for vaccine efficacy (favouring men; ref: https://www.mdpi.com/2076-393X/9/8/825). Please discuss.

8. Pay attention on the grammatical errors.

Reviewer #3: The authors provide an article on the important topic of SARS-CoV-2 seroprevalence in children during the COVID-19 pandemic in Thailand. The authors estimated infection-induced seroprevalence considering pre- and Omicron waves and reported factors associated with seropositivity.

Although the manuscript addresses a relevant topic and has some strengths, there are major statistical and presentational issues, outlined below.

1. To estimate the seroprevalence of SARS-CoV-2 the authors relied on a longitudinal study of pertussis vaccine immunity. The serum samples were collected from the participants between January 1, 2021, through November 9, 2022, in small amounts each month. Seroprevalence of SARS-CoV-2 was estimated for the overall study period, pre-omicron period, omicron period, and on a quarter and monthly basis. The method description is incomplete and inconsistent with the results section. Quarter/Monthly estimates are shown in figure 2 and figure 3 but are not mentioned in the methods section. A number of samples collected monthly and 95% confidence intervals for monthly/quarterly estimates also are not reported.

2. Figure 2 is too busy and difficult to follow. It has two Y axis, on the right side of the graph the authors present the number of positive samples, and the scale range is from 0 to 800, it is not clear if the bars represent a number of positive samples or infection-induced seroprevalence. Also, only 438 samples were collected during the study, why axis limit is established at 800?

3. As it follows from the description of the method, most of the participants were followed up and observed twice, but some of them were new recruitments during the omicron wave. This inclusion strategy resulted in two partially overlapping groups, one observed in pre-Omicron and another in the Omicron predominance period). The authors performed several comparisons of proportions, and sometimes comparisons presented in Table 1 are made between overlapping groups. The authors used the chi-square test for all comparisons, however, the use of the classical chi-square test does not seem appropriate when the groups are overlapping.

4. The authors also aimed to identify Factors associated with infection-induced seropositivity and reported risk ratios. The risk ratio is a measure of association used in cohort studies. In a cohort design, all the participants should be free of the outcome at baseline. According to the information presented by the authors, it is not the case, some of the participants were seropositive at recruitment, so the risk ratio does not seem to be an appropriate measure of association in this particular case.

Reviewer #4: The researchers conducted a longitudinal study to examine SARS-CoV-2 infection-induced seroprevalence in children and to estimate the risk variables for seropositivity. Anti-nucleocapsid (N) IgG and anti-receptor binding domain (RBD) IgG were identified using a chemiluminescent microparticle immunoassay (CMIA), and anti-RBD Immunoglobulin (Ig) was detected using an electrochemiluminescence immunoassay (ECLIA). The study addresses an important issue and has performed good comparison of effect of vaccination vs natural infection.

This reviewer has some concerns and suggestions for the authors to consider:

Specific comments:

1. Line 79: In the introduction section, it is important to mention in the manuscript regarding which vaccines are inactivated vaccines, which ones are mRNA vaccine etc to enable the understanding and relevance of this sentence and the overall rationale of this study. Same applies to the BNT162b2 vaccine described in this study.

2. Lines 85-91: These sentences need to be rephrased to enable understanding of their implied meaning. In addition, the Comment 1 is also relevant to make the rationale of the study.

3. Lines 188-191: This section needs to be re-phrased properly.

4. Was there any correlation of anti-RBD antibody levels with disease severity? Please mention it in the Results and Discussion section.

5. Why is just IgG employed for anti-receptor binding domain? Please explain the choice with supporting reference. Please also mention the limitations/disadvantages of using IgM based detection?

6. Please mention in the manuscript (with supporting references) if the anti-RBD IgG has neutralising potency.

7. Generalised comment: The language of manuscript needs major improvement. There are several grammatical mistakes. Please have it fixed.

6. PLOS authors have the option to publish the peer review history of their article (what does this mean?). If published, this will include your full peer review and any attached files.

Reviewer #1: **Yes: **Dr. Gaurav Raj Dwivedi

Reviewer #2: **Yes: **Muhammad Iqhrammullah

Reviewer #3: No

Reviewer #4: No

---

## [Author Response · Author response to Decision Letter 0]

23 Mar 2023

Response to reviewers 

Reviewer #1: 

This study by Suntronwong et al describes “SARS-CoV-2 infection- induced seroprevalence among children and associated risk factors during pre- and omicron-dominant wave, from January 2021 through November 2022, Thailand: Longitudinal study”. It will be nice study if the authors will consider few points and answer it properly:

Ans: Thank you for your valuable comments and suggestions. Please see our response below.

Abstract

1. At end of abstract, authors should emphasis on novelty and impact of the study.

Ans: This has been done.

Introduction:

I. What was the criterion for selection of pediatric population?

Ans: The inclusion criteria were healthy children who were followed up in an immunogenicity cohort study between 2015-2022 [Wanlapakorn N et al., 2020]. Children aged between 5-7 years old whose parents were able to provide written informed consent were enrolled. Exclusion criteria were parents who did not want to disclose vaccination and infection history and children who previously received any dose of the inactivated COVID-19 vaccine, as they could elicit anti-N IgG from the inactivated vaccine. These sentences have been revised in lines 131-136.

I. Is any similar type of data was reported from any part of the world. If yes, then please include in introduction?

Ans: Yes, there were a few studies that reported infection-induced seroprevalence in Uganda, U.S., and South Africa. We are now addressed in lines 107-111.

II. Authors are also suggested to cite the some of the recently published papers regarding the longitudinal study regarding SARS-CoV-2 infection- induced seroprevalence.

Ans: This has been done in lines 107-108.

Result and discussion

• Why the samples were taken from Center of Excellence in Clinical Virology, Chulalongkorn Memorial Hospital, Bangkok, Thailand.

Ans: In this study, we enrolled children who were followed up in the longitudinal serological study of pertussis vaccine immunity at the clinical trial unit of the Center of Excellence in Clinical Virology at Chulalongkorn Memorial Hospital in Bangkok, Thailand. Therefore, sera samples were taken from children in the clinical trial and tested for SARS-CoV-2 antibodies at the laboratory of the same center.

• What was the control for this study?

Ans: There is no control group. The study design focus on comparing the seroprevalence difference between the pre- and omicron-dominant wave of COVID-19 pandemic in a longitudinal cohort study. 

• Why these particular samples were taken?

Ans: In the longitudinal cohort of immunological study, the protocol stated that children undergo an annual follow-up visit to monitor their pertussis childhood vaccine-induced immunity. Left-over serum samples were tested for SARS-CoV-2 antibodies. Paired serum samples were collected from children who were followed up for one year, with the time of blood collection corresponding to the pre-omicron wave (January-December 2021) and the omicron-dominant wave (January -December 2022). This comparison between the two time periods help to determine the different in seroprevalence rate.

• What is the importance of this study as the samples were taken from specific area?

Ans: This study was conducted in children residing in Bangkok during the pre- and omicron-dominant waves. This data represented the seroprevalence in Bangkok, Thailand.

• Why the samples were taken in between January and December?

Ans: Omicron was first detected in Thailand in mid-December 2021. The incidence of omicron infection rapidly increased and subsequently became the predominant variant by January 2022. Therefore, the samples that were collected between January-December 2021 represent the pre-omicron wave, while the samples that were collected between January-December 2022 represent omicron-dominant wave.

• As the time duration is too long where there is possibility of change in sero-positivity.

Ans: We agree with your comment and mention this limitation in lines 313-318. However, we also detected and used anti-RBD IgG or anti-N IgG in unvaccinated children (all children enrolled during the pre-omicron wave had never received any COVID-19 vaccine) to represent the infection-induced seropositive. As we know that the anti-N IgG was not long-lasting immunity, our study included anti-RBD antibodies which persisted longer (391-419) days after infection to overcome this issue. 

Conclusion

Please add impact of study on the national level.

Ans: We now add the impact of this study in lines 321-323.

Reviewer #2: 

1. Abstract. Briefly present the inclusion/exclusion criteria in the ‘method’ part. Children age range should be disclose too.

Ans: This has been done.

2. In introduction. Authors are encouraged to provide the Omicron prevalence among children.

Ans: This has been done in lines 104-106.

3. In introduction. Please include the necessity of performing this study especially in the midst of global emergence of Omicron VOC (https://www.sciencedirect.com/science/article/pii/S1876034122003161?via%3Dihub#fig0010)

Ans: We now mention in lines 274-275.

4. Please explain the impact of low seroconversion among CoronaVac recipients to the results of your study. (Ref: https://www.narraj.org/main/article/view/71)

Ans: We excluded the CoronaVac-vaccinated children from this study.

5. High population burden of Omicron variant (B.1.1.529) has been associated with the emergence of severe hepatitis of unknown etiology in children (ref: https://doi.org/10.1016/j.ijid.2022.05.028) How this finding is relevant to your study?

Ans: We do not have a case of severe hepatitis of unknown etiology in children reported in Thailand. So, we cannot estimate the association between the prevalence of severe hepatitis and the seroprevalence of micron infection in children.

6. There is a good report titled “The race for clinical trials on Omicron-based COVID-19 vaccine candidates: Updates from global databases” (https://www.narraj.org/main/article/view/88) which authors may wish to incorporate in their discussion.

Ans: We now mention this point in lines 274-278.

7. It is interesting that there is no meaningful difference on the infection-induced seroprevalence between boy and girl. Meanwhile, gender has been an associated factor for vaccine efficacy (favouring men; ref: https://www.mdpi.com/2076-393X/9/8/825). Please discuss.

Ans: This has been done in lines 285-289.

8. Pay attention on the grammatical errors.

Ans: This has been done.

Reviewer #3: 

The authors provide an article on the important topic of SARS-CoV-2 seroprevalence in children during the COVID-19 pandemic in Thailand. The authors estimated infection-induced seroprevalence considering pre- and Omicron waves and reported factors associated with seropositivity.

Although the manuscript addresses a relevant topic and has some strengths, there are major statistical and presentational issues, outlined below.

Ans: Thank you for your suggestions. Please see our response below.

1. To estimate the seroprevalence of SARS-CoV-2 the authors relied on a longitudinal study of pertussis vaccine immunity. The serum samples were collected from the participants between January 1, 2021, through November 9, 2022, in small amounts each month. Seroprevalence of SARS-CoV-2 was estimated for the overall study period, pre-omicron period, omicron period, and on a quarter and monthly basis. The method description is incomplete and inconsistent with the results section. Quarter/Monthly estimates are shown in figure 2 and figure 3 but are not mentioned in the methods section. A number of samples collected monthly and 95% confidence intervals for monthly/quarterly estimates also are not reported.

Ans: We now revised the method in lines 165-167. We have reported the monthly collection of samples in Table S2. 

2. Figure 2 is too busy and difficult to follow. It has two Y axis, on the right side of the graph the authors present the number of positive samples, and the scale range is from 0 to 800, it is not clear if the bars represent a number of positive samples or infection-induced seroprevalence. Also, only 438 samples were collected during the study, why axis limit is established at 800?

Ans: We have revised the legend of Figure 2 to clarify that the right y-axis represents the number of nasopharyngeal samples identified as SARS-CoV-2 variants by sequencing or multiplex real-time reverse transcription polymerase chain reaction (RT-PCR), with a scale established at 800. These data obtained from our previous study (Puenpa J et al., 2023). 

3. As it follows from the description of the method, most of the participants were followed up and observed twice, but some of them were new recruitments during the omicron wave. This inclusion strategy resulted in two partially overlapping groups, one observed in pre-Omicron and another in the Omicron predominance period). The authors performed several comparisons of proportions, and sometimes comparisons presented in Table 1 are made between overlapping groups. The authors used the chi-square test for all comparisons, however, the use of the classical chi-square test does not seem appropriate when the groups are overlapping.

Ans: We have included the data for December 2022 and selected only individuals who were followed up from January -December 2021. Additionally, we excluded eight individuals who were newly recruited during omicron-dominant wave, resulting in two overlapping group for analysis. Therefore, we analyzed the odds ratio using McNemar test for paired serum samples, comparing between pre- and omicron dominant wave, and the chi-square test for independent samples. The method has been revised in lines 168-170.

4. The authors also aimed to identify Factors associated with infection-induced seropositivity and reported risk ratios. The risk ratio is a measure of association used in cohort studies. In a cohort design, all the participants should be free of the outcome at baseline. According to the information presented by the authors, it is not the case, some of the participants were seropositive at recruitment, so the risk ratio does not seem to be an appropriate measure of association in this particular case.

Ans: We have revised Table 1 and the statistical analysis in lines 168-170 by using the odds ratio to examine the association between seropositive and potential risk factors, as several previous reports have done, instead of using the risk ratio.

Reviewer #4: 

The researchers conducted a longitudinal study to examine SARS-CoV-2 infection-induced seroprevalence in children and to estimate the risk variables for seropositivity. Anti-nucleocapsid (N) IgG and anti-receptor binding domain (RBD) IgG were identified using a chemiluminescent microparticle immunoassay (CMIA), and anti-RBD Immunoglobulin (Ig) was detected using an electrochemiluminescence immunoassay (ECLIA). The study addresses an important issue and has performed good comparison of effect of vaccination vs natural infection.

Ans: Thank you for your suggestions. Please see our response below.

This reviewer has some concerns and suggestions for the authors to consider:

Specific comments:

1. Line 79: In the introduction section, it is important to mention in the manuscript regarding which vaccines are inactivated vaccines, which ones are mRNA vaccine etc to enable the understanding and relevance of this sentence and the overall rationale of this study. Same applies to the BNT162b2 vaccine described in this study.

Ans: We now mention in line 80-81.

2. Lines 85-91: These sentences need to be rephrased to enable understanding of their implied meaning. In addition, the Comment 1 is also relevant to make the rationale of the study.

Ans: These sentences have been revised in line 87-92.

3. Lines 188-191: This section needs to be re-phrased properly.

Ans: This sentence has been revised in lines 206-209.

4. Was there any correlation of anti-RBD antibody levels with disease severity? Please mention it in the Results and Discussion section.

Ans: Yes, a previous study has shown that individuals with disease severity could elicit a high titer of anti-RBD antibodies (Park JH et al., 2022 https://doi.org/10.1016/j.jmii.2022.04.005). However, since all children enrolled in this study presented with either asymptomatic or mild disease, we did not analyze the correlation between anti-RBD antibody levels and disease severity.

5. Why is just IgG employed for anti-receptor binding domain? Please explain the choice with supporting reference. Please also mention the limitations/disadvantages of using IgM based detection?

Ans: During the first week following SARS-CoV-2 infection, IgM levels increased and peaked for two weeks before rapidly decreasing to near-background levels in most patients. On the other hand, IgG levels were detectable after the first week and seemed to decrease more slowly over time compared to other types of antibodies (Hou H et al.,2020). A previous study has indicated that anti-Spike IgG antibodies remain seropositive for 382-491 days after infection in unvaccinated children who have been infected (Seery et al., 2022). Therefore, assays that measure total antibodies or IgG may be more sensitive indicators than IgM assays as more time passes since the person’s last infection. As the participants in this study were monitored annually, IgG was selected as a marker to detect immunity induced by either infection or vaccination in the present study.

6. Please mention in the manuscript (with supporting references) if the anti-RBD IgG has neutralising potency.

Ans: We now mention in line 94-97.

7. Generalised comment: The language of manuscript needs major improvement. There are several grammatical mistakes. Please have it fixed.

Ans: This has been done.

---

## [Decision Letter · Decision Letter 1]

16 Apr 2023

SARS-CoV-2 infection- induced seroprevalence among children and associated risk factors during the pre- and omicron-dominant wave, from January 2021 through December 2022, Thailand: A longitudinal study

PONE-D-22-32927R1

Dear Dr. Poovorawan,

We’re pleased to inform you that your manuscript has been judged scientifically suitable for publication and will be formally accepted for publication once it meets all outstanding technical requirements.

Kind regards,

Harapan Harapan, MD, PhD

Academic Editor

PLOS ONE

Additional Editor Comments (optional):

Reviewers' comments:

Reviewer's Responses to Questions

**Comments to the Author**

1. If the authors have adequately addressed your comments raised in a previous round of review and you feel that this manuscript is now acceptable for publication, you may indicate that here to bypass the “Comments to the Author” section, enter your conflict of interest statement in the “Confidential to Editor” section, and submit your "Accept" recommendation.

Reviewer #2: All comments have been addressed

Reviewer #4: All comments have been addressed

2. Is the manuscript technically sound, and do the data support the conclusions?

Reviewer #2: Yes

Reviewer #4: Yes

3. Has the statistical analysis been performed appropriately and rigorously? 

Reviewer #2: Yes

Reviewer #4: I Don't Know

4. Have the authors made all data underlying the findings in their manuscript fully available?

Reviewer #2: Yes

Reviewer #4: Yes

5. Is the manuscript presented in an intelligible fashion and written in standard English?

Reviewer #2: Yes

Reviewer #4: Yes

6. Review Comments to the Author

Reviewer #2: In general, authors have address all my previous concern, thereby can be accepted for publication. Nevertheless, I suggest author to include the justification of the time span for sera collection "from January 1, 2021

to December 14, 2022 ". Reasons for date selection should be disclosed in the method. Moreover, authors did not recruit control group, and this can be added as limitation of the study.

Reviewer #4: The revised version of the manuscript entitled "SARS-CoV-2 infection- induced seroprevalence among children and associated risk factors during the pre- and omicron-dominant wave, from January 2021 through December 2022, Thailand: A longitudinal study" can be accepted.

7. PLOS authors have the option to publish the peer review history of their article (what does this mean?). If published, this will include your full peer review and any attached files.

Reviewer #2: No

Reviewer #4: No

---

## [Editor Report · Acceptance letter]

19 Apr 2023

PONE-D-22-32927R1 

SARS-CoV-2 infection- induced seroprevalence among children and associated risk factors during the pre- and omicron-dominant wave, from January 2021 through December 2022, Thailand: A longitudinal study 

Dear Dr. Poovorawan:

I'm pleased to inform you that your manuscript has been deemed suitable for publication in PLOS ONE. Congratulations! Your manuscript is now with our production department. 

Kind regards, 

on behalf of

Dr. Harapan Harapan 

Academic Editor

PLOS ONE